# Three Biannual Rotations Cycles with Residue Incorporation Affect Wheat Production and Chemical Soil Properties

**DOI:** 10.3390/plants12244194

**Published:** 2023-12-18

**Authors:** Juan Hirzel, Pablo Undurraga, Carola Vera, Iván Matus, Pascal Michelow

**Affiliations:** 1Instituto de Investigaciones Agropecuarias, INIA Quilamapu, Av. Vicente Méndez 515, Chillán 3800062, Chile; pundurra@inia.cl (P.U.); carola.vera@inia.cl (C.V.); imatus@inia.cl (I.M.); 2Semillas Generación 2000, Carlos Sanhueza 35, Vilcún 4880045, Chile; pmichelow@uc.cl

**Keywords:** crop production, soil fertility, soil conservation practices, carbon recycling

## Abstract

Background: There are few reports of crop rotations with high residue incorporation in terms of their effects on indicator crop yields and soil properties, so this study evaluated the effect of two medium-term biannual rotations on wheat yield development and soil chemical properties after six years of rotation. Methods: The experiment was conducted with two biannual rotations (canola–wheat and bean–wheat) and four residue incorporation levels (0%, 50%, 100%, and 200%) in an Andisol in south central Chile. Wheat grain yield and residue production were evaluated during each biannual cycle during three cycles of crop rotation, and soil chemical properties were evaluated at final evaluation. Results: The use of beans as a wheat preculture partially improved grain yield in 7.3%. The chemical properties of the soil showed an increase in pH (0.08 units), organic matter content (15 g kg^−1^), and concentrations of P (2.8 mg kg^−1^), S (7.4 mg kg^−1^), and Al (0.03 cmol_+_ kg^−1^) after canola cultivation, while after bean cultivation there was an increase in the available N concentration (3.7 mg kg^−1^). The use of increasing doses of residue allowed for an increase in the soil pH and decrease in the exchangeable Al concentration. Conclusion: The continuous incorporation of the residues produced within the biannual rotations evaluated in this volcanic soil did contribute to improving some chemical properties of the soil without affecting wheat crop yield.

## 1. Introduction

Worldwide grain production is of utmost importance for basic food needs, with wheat (*Triticum aestivum* L.), rice (*Oryza sativa* L.), and maize (*Zea mays* L.) crops being the most prominent in terms of surface area [1,2]. Considering that the available surface area for extensive crops is limited worldwide, over time, agronomic practices have been generated and implemented to increase the productivity of these crops, which in turn generates greater residue productivity [3,4,5]. The generated residues can have many uses, such as agricultural purposes for nutrient recycling (carbon (C), nitrogen (N), and other macronutrients) and improving soil organic matter content (SOM) and soil fertility [6,7,8,9], animal feed [10], and also for energy generation, such as biofuels [11,12]. Additionally, residue incorporation helps mitigate the greenhouse gas effect by reducing CO_2_ emissions [13]. Regarding the use of crop residues in agriculture, there are still many questions, like possible negative effects on nutrient availability such as N due to microbial organization processes [11,12,14,15], and physical limitations for soil preparation and the establishment of the following crop within a rotation [16]. In turn, one of the practices that has contributed to facilitating the use of agricultural residues is precisely the use of crop rotations, which allows for time windows to be left between the incorporation of residues from the recently harvested crop and the sowing of the next crop, as well as improving microbial biodiversity in the soil, which increases the natural capacity of the soil system to achieve residue decomposition within the soil [11,17].

Wheat production in Central and Latin America covers an area of 9.1 million ha with an average yield of 3.2 Mg ha^−1^ [1], while for Chile, this area is 173,106 ha with an average yield of 5.8 Mg ha^−1^ [18], which generates attention to the proper use of residues produced by this crop. Several authors have shown the beneficial effects of incorporating crop residues on soil properties [6,8,12,14] and on the productivity of the following crop, especially for rotation conditions using legumes [7,14,19,20,21]. Chen et al. [6] in a six-year study with biennial corn-wheat rotations indicated that residue incorporation led to an increase in carbohydrate-derived components of SOM (carboxylic acid groups and/or esters, amides, and aromatic compounds) and an increase in organic C contents, total N, ammoniacal N, and C/N ratio in the soil compared to the control without residue application. Kumar et al. [8], working with residue incorporation from five crop rotations, fallow–rice–rice, jute (*Manihot esculenta* Crantz)–rice–wheat, jute–rice–baby corn, jute–rice–garden pea (*Pisum sativum* L.), and jute–rice–mustard (*Sinapsis alba* L.)–mung bean (*Vigna radiata* L.)/green gram (*Vigna radiata* L.) (residue of rice, wheat, and corn at 4 Mg ha^−1^ and garden pea and mung bean at 2 Mg ha^−1^ incorporated into soil with their respective cropping), with two fertilization levels (75% and 100% of recommended doses), found greater mineralization of C and N in rotations that included legumes, and that higher fertilization doses also increased this mineralization. Basir et al. [14] indicated that corn residue incorporation with surface tillage and nitrogen fertilization induced an increase in wheat grain yield and at the same time improved the physicochemical properties of the soil compared to the treatment without residue incorporation or with residue burning.

While the literature cited in this and other articles refers to benefits of residue incorporation on soil properties, few studies indicate the effect of using high doses of residues on soil properties and on the yield of indicator crops. On the other hand, in Central and Latin America, there is an important area cropped with wheat on different types of soils, among which we can find volcanic soils with a high organic matter content, which may be a reason why farmers do not consider it necessary to add organic matter to the soil. Considering this background, our working hypothesis is that crop rotations with residue incorporation in the medium- to long-term can improve the chemical properties of volcanic soils and probably contribute to an increase in wheat yield. Considering that wheat production in an important area of Latin America does not include legumes and oilseeds, among which beans stand out for their nutritional importance and canola for its agro-industrial importance, which could also contribute to improving soil properties or increasing wheat crop yields within the rotation through the incorporation of their residues (eliminating the burning of these residues), the present study evaluated the effect of three cycles of two biannual rotations, including bean and canola as a pre-crop (bean–wheat and canola–wheat) and four residue incorporation levels for each crop (0%, 50%, 100%, and 200%) on wheat yield, residue production, and chemical properties at the end of the evaluation period.

## 2. Materials and Methods

The experiment was conducted for six consecutive seasons from 2016 to 2022 at the Santa Rosa Experimental Station, INIA-Quilamapu, Chillán, Chile (36°31′ S; 71°54′ W). The soil is volcanic (Melanoxerand) with a moderate effective depth (0.45 to 0.60 m) and the climate is temperate Mediterranean characterized by a hot, dry summer and cold, wet winter. Precipitation was 605, 563, 730, 460, 576, and 920 mm for the 2016–2017, 2017–2018, 2018–2019, 2019–2020, 2020–2021, and 2021–2022 seasons, respectively, which was concentrated in winter and spring. The mean temperature was 12.8, 13.2, 13.5, 13.4, 14.3, and 13.2 °C, and evaporation was 1023, 1041, 990, 980, 1060, and 966 mm for the 2016–2017, 2017–2018, 2018–2019, 2019–2020, 2020–2021, and 2021–2022 seasons, respectively.

### 2.1. Experiment Management

The design of this long-term experiment consisted of biannual rotations combining two crops, bean–wheat and canola–wheat, in which residues of the previous crop have been incorporated at levels of 0%, 50%, 100%, and 200%; the basic design has been maintained over time. The present article focuses on grain yield and residue production of the wheat as the second crop in each biannual rotation (2017–2018; 2019–2020; and 2021–2022 season), and on the soil chemical properties at the end of the evaluation period (2022). Lime was applied at the rate of 3,000 kg ha^−1^ before the start of the biannual rotations in April 2016 to correct soil acidity (Table 1). There were two previous crops before the wheat crop: (1) canola (*Brassica napus* L.) crop and (2) bean (*Phaseolus vulgaris* L.) crop. The experimental unit for each crop rotation (bean–wheat and canola–wheat) had a 40 m long and 14 m wide (560 m^2^) plot with 0.7, 0.7, and 0.2 m inter-row spacing for the bean, canola, and wheat crops, respectively, and the plot was divided into four split plots that were 20 m long and 7 m wide (140 m^2^) for the incorporation of each residue level (0, 50, 100, and 200%). So, the total experimental area was 4,480 m^2^ and included two crop rotations and four replicates.

The canola crop in the first rotation cycle was sown on 15 May 2016 and harvested on 5 February 2017, while it was sown on 25 May 2018 and harvested on 15 January 2019 in the second rotation cycle, whereas for the third season, it was sown on 31 August 2020 and harvest on 25 January 2021. Canola ‘Eminem-von Baer’ was used in the first two seasons, and ‘Imminent-SIS’ (a new hybrid with better yield potential for the study area) was used in the third season, and the seed rate in each season was 30 kg·ha^−1^. Irrigation was applied at the flowering stage. Total weed control was carried out with the herbicide propisochlor (Proponit 720 EC) at 1.44 kg a.i.·ha^−1^, and disease control was not necessary. Nitrogen, P (P_2_O_5_), and K (K_2_O) fertilization rates were 160, 120, and 80 kg ha^−1^, respectively. Both P and K were applied 100% at sowing, while N was applied 50% at sowing and the remaining 50% was applied at the 60% crop cover stage. Fertilizer sources were urea, triple superphosphate, and potassium chloride. In addition, Mg, S, Zn, and B were applied at rates of 30:33:4:2 kg ha^−1^ before sowing with magnesium sulfate, zinc sulfate, and calcium borate fertilizers based on soil chemical properties (Table 1).

The bean crop ‘Zorzal-INIA’ was sown on 27 October 2016 and harvested on 28 February 2017 in the first rotation cycle, while the second rotation cycle was sown with ‘Torcaza-INIA’ (a new cultivar with a better yield potential for the study area) on 27 October 2018 and harvested on 28 February 2019, whereas for the third season ‘Torcaza-INIA’ was sown on 10 November 2020 and harvested on 11 March 2021. The seed rate was 120 kg·ha^−1^ in the three seasons. Irrigation was applied at the flowering stage. Total weed control was carried out with the herbicide fomesafen (Flex: 25%) at 0.25 kg a.i.·ha^−1^ at the first trifoliate leaf, and disease control was not necessary. The N, P (P_2_O_5_), and K (K_2_O) fertilization rates were 60, 60, and 60 kg ha^−1^, respectively. Nitrogen, P, and K were applied 100% at sowing, and fertilizer sources were urea, triple superphosphate, and potassium chloride. In addition, Mg, S, Zn, and B were applied at rates of 30:33:4:2 kg ha^−1^ before sowing with magnesium sulfate, zinc sulfate, and calcium borate fertilizers based on soil chemical properties (Table 1).

The wheat crop ‘Pandora-INIA’ was sown on 15 July 2017 and harvested on 20 January 2018 in the first rotation cycle, while it was sown on 10 July 2019 and harvested on 22 January 2020 in the second rotation cycle, whereas it was sown on 12 July 2021 and harvested on 15 January 2022 in the third rotation cycle. The seed rate was 220 kg·ha^−1^ in the three seasons. Irrigation in wheat was applied at the booting, heading, and milk to dough stages. Total weed control was carried out with Propisochlor (Proponit 720 EC) at 432 g a.i.·ha^−1^ and flumiozaxin (Pledge 50 WP) at 25 g a.i.·ha^−1^ at the pre-sowing stage, tritosulphuron at 50 g a.i.·ha^−1^ with dicamba at 100 g a.i.·ha^−1^ (Arrat), metsulfuron-methyl (Aliado) at 4.8 g a.i.·ha^−1^ at the tillering stage, and disease control was not necessary. Nitrogen, P (P_2_O_5_), and K (K_2_O) fertilization rates were 240, 120, and 120 kg ha^−1^, respectively, based on soil chemical properties (Table 1). Phosphorous and K were applied 100% at sowing, while N was applied 15%, 45%, and 40% at the sowing, tillering, and flag leaf stages. Fertilizer sources were urea, triple superphosphate, and potassium chloride. Based on the soil chemical analysis (Table 1), Mg, S, Zn, and B were applied at rates of 30:33:4:2 kg ha^−1^ before sowing in all crops with magnesium sulfate, zinc sulfate, and calcium borate fertilizers.

Once the three crops were harvested, residues were incorporated at levels of 0%, 50%, 100%, and 200% during May in each year in the same experimental unit. The machinery used to grind and incorporate residues was a displaceable mulcher (Tornado 310, Maschio Gaspardo, Campodarsego, Italy) and a compact disk harrow (Rubin 9, Lemken GmbH and Co. KG, Alpen, Germany), respectively.

### 2.2. Wheat Yield and Residue Production

The plots were harvested manually at grain maturity and threshed with a stationary thresher. Plant samples were collected from a 2.1 m^2^ plot area and separated as grain and aerial residue. Grain and tissue samples were oven-dried at 70 °C for 72 h.

### 2.3. Soil Analysis

At the beginning of the experiment, composite samples were collected manually from the 0 to 20 cm soil depth for each treatment on the same day that the canola crop was harvested. All samples were air-dried and sieved (2 mm mesh). Soil pH was determined in 1:2.5 soil/water extracts. Soil organic C was established using Walkley–Black wet digestion [22]. Soil inorganic N (NO_3_-N and NH_4_-N) was extracted with 2 M KCl and determined by colorimetry with a segmented flux spectrophotometer (autoanalyzer, Skalar Analytical BV, Breda, the Netherlands). Soil extractable P was 0.5 M NaHCO_3_ (Olsen P) using the molybdate-ascorbic acid method. Exchangeable Ca, Mg, K, and Na were determined by 1 M NH_4_OAc extraction followed by flame spectroscopy: absorption (Ca and Mg) and emission (K and Na). The soil-exchangeable Al concentration was obtained with 1 M KCl extraction via absorption spectroscopy. Sulfur (SO_4_^2−^) was determined with calcium phosphate 0.01 M and turbidimetry. Ending season 2021–2022 (May of 2022) composite samples were collected manually from the 0 to 20 cm soil depth for each treatment and replicate and were analyzed using the methodologies described above.

### 2.4. Experimental Design and Statistical Analysis

The experimental design was a split plot in which the main plot was the previous crop (two crops) and the split plot was the residue level (four levels) with four replicates. Grain yield and residue production of the wheat crop were analyzed as an effect of each two-year rotation (3 cycles), while soil chemical properties were analyzed at the end of the three rotation cycles. Results were analyzed by an ANOVA and Tukey’s test (*p* = 0.05) using the SAS PROC MIXED Model procedure [23]. For significant interactions, contrast analysis was used to compare the treatment effects separately (*p* = 0.05).

## 3. Results

The significance analysis indicated that wheat grain yield was affected by the evaluation year and the interaction between year and crop rotation (*p* < 0.05), while residue production was not affected by any of the sources of variation (*p* > 0.05) (Table 2). For grain production, the interaction between years and crop rotation only showed a difference in the first evaluation season (Figure 1), where wheat grain yield was 7.3% higher after bean cultivation compared to the use of canola as a preculture (*p* < 0.05). On average, for each evaluation year and compared to the value obtained for the first year (5.94 Mg ha^−1^), grain yield was 1.3% lower in the second season (5.87 Mg ha^−1^) and 6.4% higher in the third season (6.32 Mg ha^−1^) (Figure 1), but the mean test employed did not detect significant differences for the average yield between evaluation seasons. Residue production fluctuated between 6.83 and 7.36 Mg ha^−1^.

The chemical properties of the soil in the first 20 cm of depth at the end of the evaluation period indicated that crop rotation significantly affected the pH, organic matter content, the concentrations of available N, P, and S, and the exchangeable Al, while the residue level affected the pH and the concentrations of both exchangeable Mg and Al (Table 3). As an average of the four residue levels used, the use of canola as a pre-crop for wheat induced an increase in the pH, organic matter level, the concentrations of both available P and S, and the exchangeable Al (*p* < 0.05), while using bean as a preculture increased the available N (*p* < 0.05) (Table 4). As an average of both crop rotations, the incorporation of increasing doses of residues had a directly proportional effect on the increase in soil pH and an inversely proportional relationship on the concentration of exchangeable Al (Table 5); however, the correlation values were low and corresponded to 0.43 for soil pH-residue level and 0.35 for exchangeable Al level. The effect of the residue level on soil-exchangeable Mg was erratic, and a higher concentration was only observed compared to the control when the highest dose of residue was used (*p* < 0.05) (Table 5).

## 4. Discussion

The wheat grain yield and its residue production were normal for the study area [24]. Although there were differences in rainfall between the study seasons that may have affected yield, for those seasons with lower rainfall, more supplementary irrigation was carried out, which provided adequate growing conditions for the crop. The beneficial effect of using bean as a pre-crop on grain production was only observed in the first evaluation season of this experiment, probably due to the high fertilization used in wheat cultivation, which decreases the response to the greater natural delivery of nutrients when considering legumes in crop rotation [5,25,26]. The different levels of incorporated residue did not affect the grain or residue production of the wheat crop, unlike what was indicated by some authors [8,14,20,27,28]. This can be explained by biological processes associated with the entry of C into the soil, which allows for the formation of highly organized organic compounds [6,15,29,30,31], though these were not evaluated in this experiment.

Regarding the chemical properties of the soil at the end of the three biennial rotation cycles (6 years after the start of the experiment) and in relation to the beginning of the experiment, an increase in pH and concentrations of both exchangeable Ca and Mg was observed. Also, there was a decrease in both exchangeable Na and Al in response to liming (CaCO_3_*MgCO_3_) carried out at the beginning of the experiment [30,32]. The concentration of available S also increased compared to the initial value, which may respond to the fertilization used during the six years of cultivation. The use of canola as a pre-crop induced an increase in soil pH, organic matter content, and concentrations of both available P and S, as well as exchangeable Al, which may be associated with the greater mass of residue incorporated with canola that allows for greater recycling of the amount of C and mineral nutrients [8,30,33]. Canola residue incorporation to the soil has been reported to increase available K [34] and S concentrations [35,36,37] because of the high concentration of these nutrients in stems and siliques and their lower C:S ratio compared with other crops [38]; moreover, exchangeable K was not observed to have an effect in our experiment. Canola residue presents a different organic composition than bean (polyphenols, polyphenols:N ratio, and lignin content), which affects the organic matter oxidation and the mineralization rate by the soil biomass, generating a greater accumulation of soil organic matter and a lower available N concentration [30,38,39]. The use of bean as a catch crop increased the concentration of available N, which has been described by other researchers as a positive effect of legume use [5,40,41]. In this regard, Woźniak [41] indicated that the inclusion of legumes in crop rotation in Poland increased organic C and total soil N, relative to the use of raps, which was partly explained by higher soil biological activity. Thus, the soil N concentration obtained in rotations, including beans, was higher than the value in oilseed rape–wheat rotations.

The use of increasing doses of residue induced an increase in soil pH, associated with the recycling of basic reaction nutrients [30,32], and a nutritional extraction that was an expected constant given that grain yields and residue production were not affected by the increasing dose of residue. For this same experiment, similar nutrient extractions for the wheat crop were previously reported against increasing doses of residue [42]. In contrast to these results, Basir et al. [14] indicated a decrease in soil pH as an effect of the incorporation of residue, compared to the control without the use of residue, as an effect of the release of acid-reactive carbon compounds derived from soil microbial activity. In agreement with what was pointed out by Basir et al. [14], the OM content of the soil was also increased with the application of the residue, and in this case, with a relationship directly proportional to the dose of residue used, which is explained by the contribution of C. The application of increasing doses of residues generally allows for a higher concentration of both P and K available to be obtained [36,43,44]. For our experiment, there was a quantitative increase in soil-exchangeable K, which was not significant, probably associated with the spatial variability of the physical–chemical properties of the soil [45,46], which was not evaluated in this experiment.

The same effect could have influenced the observed erratic increment in exchangeable Mg in the face of increasing doses of residue. For the concentration of available P, the increasing dosage also had no effect on the residue, probably associated with biological processes of formation of stable organic compounds in the soil [15,29,30,31]. Generally, when residues are incorporated, N applications are made to avoid the “N starvation” effect associated with the nutritional requirements of the soil biomass responsible for the oxidation process of the added organic C [38]. However, in our experiment, N was not applied given the high organic matter content of the soil and the residual effect of fertilization, which allows for the N needs of the soil biomass to be supplied and, as can be seen in the grain yield, obtained with increasing residue doses. At the same time, it should be considered that the chronological time between residue incorporation and sowing of the next crop in this biannual rotation fluctuated between 5 and 9 months, which has been demonstrated for Kilimanjaro andisols [47]. In addition, the presence of plants generates exudation of carbon compounds from their roots, which activates soil biomass and contributes to the mineralization of soil organic matter decomposition of previous crop residues. Finally, the decrease in the exchangeable Al concentration can be explained by the higher contribution of basic reaction nutrients generated with the increment in the dosage of incorporated residue [30,32].

In conclusion, the use of canola or bean as a wheat pre-crop only affected grain yield during one season within the three biennial evaluation cycles, with a positive effect on the bean crop, while the production of wheat residues was not affected. The chemical properties of the soil after three cycles of biennial rotation were affected by the rotation, with an increase in pH, organic matter content, and concentrations of P, S, and Al after canola cultivation, while after bean cultivation, there was an increase in the concentration of available N. The use of augmenting doses of residue allowed for an increase in the soil pH and, in turn, a decrease in the exchangeable Al concentration. The concentration of exchangeable Mg was partially increased by the level of residue incorporated. Finally, the incorporation of residues in the evaluated crop rotations is a tool to reduce or avoid the burning of crop residues in those farming systems where wheat is the dominant crop.

## Figures and Tables

**Figure 1 plants-12-04194-f001:**
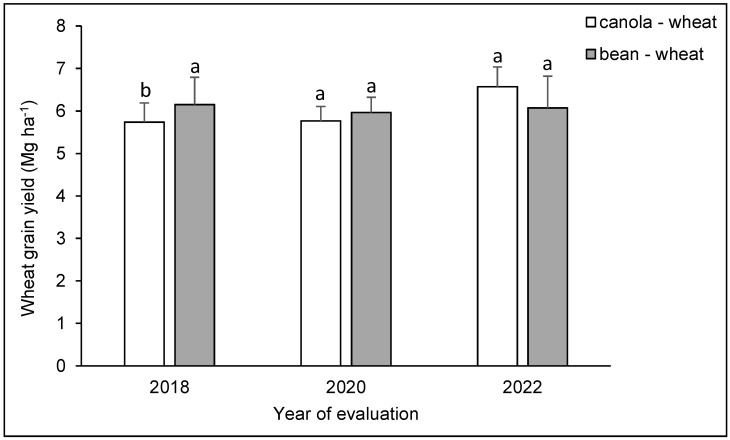
Wheat grain yield during three cycles of two biannual crop rotations (after canola or bean). Different letters above the bars indicate differences in the same year of evaluation according to Tukey’s test (*p* < 0.05). Whiskers correspond to the standard error for each bar.

**Table 1 plants-12-04194-t001:** Soil chemical properties at the 0–0.2 m soil depth before initiating the crop rotation experiment (year 2016). Methods used are described in the text.

Parameters	Value
Clay (%)	16.7
Silt (%)	44.6
Sand (%)	38.7
Bulk density (g cm^−3^)	1.00
pH _(soil:water 1:5)_	5.52
Organic matter (g kg^−1^)	109.2
EC (dS m^−1^)	0.11
Available N (mg kg^−1^)	54.1
Olsen P (mg kg^−1^)	21.3
Exchangeable K (cmol_c_ kg^−1^)	0.54
Exchangeable Ca (cmol_c_ kg^−1^)	4.20
Exchangeable Mg (cmol_c_ kg^−1^)	0.36
Exchangeable Na (cmol_c_ kg^−1^)	0.08
Exchangeable Al (cmol_c_ kg^−1^)	0.12
Available S (mg kg^−1^)	23.5

EC: electrical conductivity; N: nitrogen; P: phosphorus; K: potassium; Ca: calcium; Mg: magnesium; Na: sodium; Al: aluminum; S: sulfur.

**Table 2 plants-12-04194-t002:** Significance testing for the wheat grain yield and residue production as affected by two crop rotations with four residue incorporation levels.

Fuente de Variación	Grain Yield	Residue Production
Year (Y)	0.0068	0.68
Crop Rotation (CR)	0.74	0.25
Residue Level (RL)	0.39	0.60
Interaction Y × CR	0.0088	0.38
Interaction Y × RL	0.61	0.99
Interaction CR × RL	0.19	0.64
Interaction Y × CR × RL	0.66	0.44

**Table 3 plants-12-04194-t003:** Significance testing for the soil chemical properties as affected by two crop rotations with four residue incorporation levels after six years of evaluation.

Soil Properties	Crop Rotation (CR)	Residue Level (R)	CR × R Interaction
pH	0.028	0.043	0.90
Organic matter	<0.01	0.63	0.99
Available N	<0.01	0.06	0.55
Available P	<0.01	0.10	0.79
Exchangeable Ca	0.21	0.67	0.65
Exchangeable Mg	0.29	0.03	0.53
Exchangeable K	0.89	0.16	0.40
Exchangeable Na	0.05	0.63	0.81
Exchangeable Al	<0.01	0.02	0.20
Available S	<0.01	0.87	0.56

**Table 4 plants-12-04194-t004:** Soil chemical properties as affected by two crop rotations as an average of four residue incorporation levels.

Soil Properties	Canola–Wheat	Bean–Wheat
pH	6.02 ^a^	5.94 ^b^
OM, g kg^−1^	101.0 ^a^	86.0 ^b^
Available N, mg kg^−1^	11.5 ^b^	15.2 ^a^
Available P, mg kg^−1^	20.4 ^a^	17.6 ^b^
Exchangeable Ca, cmol_+_ kg^−1^	5.25 ^a^	4.85 ^a^
Exchangeable Mg, cmol_+_ kg^−1^	0.52 ^a^	0.49 ^a^
Exchangeable K, cmol_+_ kg^−1^	0.49 ^a^	0.49 ^a^
Exchangeable Na, cmol_+_ kg^−1^	0.06 ^a^	0.06 ^a^
Exchangeable Al, cmol_+_ kg^−1^	0.07 ^a^	0.04 ^b^
Available S, mg kg^−1^	40.2 ^a^	32.8 ^b^

Different letters in the same row indicate differences between crop rotations as an average of the four residue incorporation levels according to Tukey’s test (*p* < 0.05).

**Table 5 plants-12-04194-t005:** Soil chemical properties as affected by four residue levels as an average of two crop rotations.

Soil Properties	Residue Level (%)			
	0	50	100	200
pH	5.90 ^b^	5.96 ^ab^	6.02 ^ab^	6.04 ^a^
OM, g kg^−1^	92.0 ^a^	93.0 ^a^	93.0 ^a^	95.0 ^a^
Available N, mg kg^−1^	15.7 ^a^	13.3 ^a^	12.3 ^a^	12.1 ^a^
Available P, mg kg^−1^	19.8 ^a^	20.2 ^a^	17.8 ^a^	18.4 ^a^
Exchangeable Ca, cmol_+_ kg^−1^	5.21 ^a^	5.17 ^a^	4.72 ^a^	5.10 ^a^
Exchangeable Mg, cmol_+_ kg^−1^	0.47 ^b^	0.51 ^ab^	0.47 ^b^	0.58 ^a^
Exchangeable K, cmol_+_ kg^−1^	0.45 ^a^	0.49 ^a^	0.47 ^a^	0.55 ^a^
Exchangeable Na, cmol_+_ kg^−1^	0.07 ^a^	0.06 ^a^	0.06 ^a^	0.06 ^a^
Exchangeable Al, cmol_+_ kg^−1^	0.07 ^a^	0.06 ^ab^	0.05 ^b^	0.05 ^b^
Available S, mg kg^−1^	38.3 ^a^	35.3 ^a^	36.3 ^a^	36.2 ^a^

Different letters in the same row indicate differences between residue incorporation levels as an average of two crop rotations according to Tukey’s test (*p* < 0.05).

## Data Availability

Data of our research are contained within the article tables and figures.

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
