# Peer review of "Three Biannual Rotations Cycles with Residue Incorporation Affect Wheat Production and Chemical Soil Properties"

_plants, 2023, doi:10.3390/plants12244194_

Round 1
Reviewer 1 Report
Comments and Suggestions for Authors
The paper presents an interesting and relevant study on the effects of biannual crop rotations and residue incorporation levels on wheat yield and soil chemical properties. The paper is well-written and organized, and the methods and results are clearly described and supported by appropriate data and statistics. However, there are some aspects that could be improved to enhance the quality and clarity of the paper.
- Line 86-87 show that there was a wide range of precipitation from 2016 to 2022, especially in 2019-2020 and 2021-2022. How did this variability affect your study results and conclusions?
- Could you please explain why Figure 1 does not include the data for 2019 and 2021? What were the reasons for excluding these years from the analysis?
Author Response
Dear Reviewer 1, I would like to thank you for your suggestions and indicate the answers included in the correction made.
- Line 86-87 show that there was a wide range of precipitation from 2016 to 2022, especially in 2019-2020 and 2021-2022. How did this variability affect your study results and conclusions?. For this observation the next paragraph was include and highlighted in yellow color:
4. Discussion
The wheat grain yield and its residue production were normal for the study area [24]. Although there were differences in rainfall between the study seasons that may have affected yield, for those seasons with lower rainfall, more supplementary irrigation was carried out, which provided adequate growing conditions for the crop.
2. Could you please explain why Figure 1 does not include the data for 2019 and 2021? What were the reasons for excluding these years from the analysis?.
For both years 2019 and 2021 the crops made were canola and bean as previous crop to the Wheat crop evaluated in the season 2018, 2020 and 2022 (three cicles of biannual rotation crop).

Reviewer 2 Report
Comments and Suggestions for Authors
Review remarks on the research manuscript entitled “Three Biannual Rotations Cycles with Residue Incorporation Affect The Wheat Production and Chemical Soil Properties” submitted by Hirzel et al. to plants-MDPI journal. The authors assess the impact of two medium-term biannual rotations on wheat production and soil characteristics after six years of 10 rotations. The present article is well-written and discussed systematically.
Overall comments:
The authors have identified an important and interesting issue. Scientifically, the MS is strong, and I recommend its publication in the “plants-MDPI journal” after minor revision. English is not well presented. Even some sentences are difficult to understand.
Some specific comments are stated below:
Specific Comments:
Abstract: Correct the title ……………… Affect in place of Affecr
Abstract: Needs modification. Please include some more numerical data in the abstract.
Keywords: I suggest authors improve keywords
Introduction:
- Authors should have a strong justification for choosing 10 rotations in 6 years. Any specific reason behind this and why? It should be mentioned more elaborately in the introduction part itself.
Materials and Methods:
Please revise this section critically.
Results and Discussion: Well written but still needs more deep discussion with recent and relevant literatures.
Table 4, 5: Statistical letters should be superscript
Conclusion should also focus the key message and future scope of the study.
Check references as per guidelines
Comments on the Quality of English LanguageEnglish is not well presented. Even some sentences are difficult to understand.
Author Response
Dear reviewer 2, I would like to thank you for your suggestions and would like to indicate the corrections made, which were highlighted in green in the text.
Specific Comments:
Abstract: Correct the title ……………… Affect in place of Affecr.
The title was corrected.
Abstract: Needs modification. Please include some more numerical data in the abstract.
Numerical values was included for all the results indicate as significative difference.
Keywords: I suggest authors improve keywords.
The wors "carbon recicling" were included.
Introduction:
- Authors should have a strong justification for choosing 10 rotations in 6 years. Any specific reason behind this and why? It should be mentioned more elaborately in the introduction part itself.
The paragraph that include the selecction of two rotations was corrected and the new paragraph is the next:
Considering that wheat production in an important area of Latin America does not include legumes and oilseeds, among which beans stand out for their nutritional importance and canola for its agro-industrial importance, which could also contribute to improve soil properties or increase wheat crop yields within the rotation through the incorporation of their residues (eliminating the burning of these residues), the present study evaluated the effect of three cycles of two biannual rotations including bean and canola as a pre-crop (bean-wheat and canola-wheat) and four residue incorporation levels for each crop (0%, 50%, 100% and 200%) on wheat yield, residue production and chemical properties at the end of the evaluation period.
Materials and Methods:
Please revise this section critically.
This section was analysed, the number of years of field assessment was corrected (it previously said five and the correct number is six).
The detailed management of each crop can be very extensive and with a great degree of detail of products and doses, but the authors prefer to mention this management in case someone would like to consult the study and compare it with the specific management of another experimental area.
Results and Discussion: Well written but still needs more deep discussion with recent and relevant literatures.
The results and discussion was reviewed and in every sentence in which quotations are included, most of them are less than 10 years old, except for a few sentences in which citations are included:
32. Fageria, N.K.; Nascente, A.S. Chapter Six - Management of Soil Acidity of South American Soils for Sustainable Crop Produc-tion. In Advances in Agronomy; Sparks, D.L., Ed.; Academic Press, 2014; Vol. 128, pp. 221-275.
39. Heard, J.; Hay, D. Nutrient content, uptake pattern and carbon:nitrogen ratios of prairie crops. Manitoba Agriculture, Food and Rural Initiatives, Carman, Canada. 2006. http://umanitoba.ca/faculties/afs/MAC_proceedings/proceedings/2006/heard_hay_nutrient_uptake.pdf
Table 4, 5: Statistical letters should be superscript.
The statistical lettes were corrected.
Conclusion should also focus the key message and future scope of the study.
The conclusion was corrected and a nwe paragraph was included:
"Finally, the incorporation of residues in the evaluated crop rotations is a tool to reduce or avoid burning of crop residues in those farming systems where wheat is the dominant crop".
